# Use and Acceptance of Drinking Fountains: A Pilot Study in Two Secondary Schools in Dortmund, Germany

**DOI:** 10.3390/children10050817

**Published:** 2023-04-29

**Authors:** Martin Jakob Gerhardus, Susanne Klammer, Michael Galatsch, Ralf Weigel

**Affiliations:** 1Department of Pediatrics, University Hospital RWTH Aachen, 52074 Aachen, Germany; 2Child and Adolescent Health Services, Public Health Department Dortmund, 44137 Dortmund, Germany; 3Institute of Nursing, School of Health Science, Zürich University of Applied Science, ZHAW, 8001 Winterthur, Switzerland; 4Friede Springer Endowed Professorship for Global Child Health, School of Medicine, Faculty of Health, Witten/Herdecke University, 58455 Witten, Germany

**Keywords:** beverages, drinking, schools, health promotion, health education, drinking water, overweight, children

## Abstract

(1) Background: Water drinking is essential to reduce obesity in children, but effective means for implementation remain controversial. Our study assesses students’ and teachers’ use of and attitudes towards drinking fountains in two urban secondary schools. (2) Methods: In a cross-sectional study, answers from students and teachers to a 28- and 19-item questionnaire, respectively, containing closed- and open-ended questions and short interviews with the schools’ two principals were described and analysed using the question-specific number of responses as the denominator. (3) Results: Questionnaires of one hundred sixty-two students and ten teachers were analysed; 36.1% of students responded. Students viewed the schools’ two fountains as a good idea (73.3%, *n* = 118), recommended them to other schools (73.1%, *n* = 117), and felt able to distinguish healthy from unhealthy drinks (70.5%, *n* = 110). In contrast, 55.7% (*n* = 88) reported using the fountains regularly; over a week, 39.8% (*n* = 47) used them less than once; 26.3% (*n* = 31) used them one to two times. Only about a third (26.5%, *n* = 43) reported consuming more water since the fountains’ installation. Teachers’ responses were similar to students’; principals stressed planning and costs. (4) Conclusions: A discrepancy between a good attitude towards and actual use of drinking fountains may exist; school communities may need to look for measures to overcome it.

## 1. Introduction

Drinking sufficient water is critical for children’s diet and healthy life, but unfortunately, it is not a common practice for many children. Newborns and infants are particularly vulnerable to life-threatening dehydration [1]. Inadequate hydration negatively impacts the brain’s function and structure [2] and can even lead to chronic conditions, such as urolithiasis or bronchopulmonary disorders [3]. On the other hand, research has demonstrated that adequate water intake positively influences pupils’ cognitive abilities [4]. Moreover, water can prevent obesity when replacing caloric beverages without any other changes in the diet [5,6,7]. However, other factors, such as adopting a healthier lifestyle or increasing physical activity levels, will also play a role [8]. Surprisingly, only about one in ten schoolchildren meet the recommended water intake, as data from France suggest. Nationally representative data revealed that 90% of children did not meet the water intake recommendations of the European Food Safety Authority [9,10]. In Belgium, only 9.5% of children aged between eight and thirteen years met the recommendations [11].

In contrast, children often consume unhealthy amounts of sweetened beverages containing a significant number of calories [12]. For example, children and adolescents aged three to seventeen years drink more than two glasses of sugary drinks a day on average, and consumption increases with age, reaching a peak in the 14–17 age group, according to findings in Germany [13]. Therefore, the report’s authors stress the importance of “the continuous provision of water, unsweetened herbal or fruit teas, or highly diluted or highly diluted fruit spritzers in the educational and care facilities” in conclusion [13].

Drinking fountains could offer a promising approach to address the need for increased water consumption among schoolchildren. For instance, establishing school drinking fountains has been shown to support healthy drinking behaviour by significantly increasing students’ water consumption in several primary schools [7,14,15]. Furthermore, easy access to water through drinking fountains during school hours has been found to decrease the consumption of sweetened drinks [16]. Despite these benefits, studies conducted in primary schools in the United Kingdom (UK) and the United States (US) indicate that children still consume inadequate amounts of water, which could be attributed to negative attitudes towards tap water [17].

In Germany, the prevalence of overweight (including obesity) children aged three to seventeen years from 2014–2017 was 15.5%, with little change compared to 2003–2006. Still, obesity affected children from the lowest socioeconomic group most, and it had increased from 20% to 25.5% among them [18]. In Dortmund, the third largest city in Germany’s most populous state of North Rhine-Westphalia (NRW), 13.2% of six-year-olds were overweight in 2019, before the COVID-19 pandemic, at routine school entrance examinations. In Dortmund’s socioeconomically less privileged communities, the proportion reaches 21% [19], beyond NRW’s average of 10.7% [20], and public health authorities identified unhealthy diets as a leading cause. In response, Dortmund’s Public Health Department launched the “Iss ok” project [21] in 2016 to encourage students to adopt healthier eating and drinking habits and to replace sugary drinks in schools and at home. “Iss ok” installed drinking fountains in Dortmund’s schools; one in a municipal “Hauptschule” (general secondary school) and one in a “Gymnasium” (roughly equivalent to a UK grammar school). The pilot aimed to promote water consumption in students beyond primary school age in years 6 to 10, and 6th-year students received a recyclable plastic bottle for a refill.

This exploratory study assesses the use and acceptance of drinking fountains in the two secondary schools addressing students and teachers. Specifically, the study examines the frequency of the fountains’ use, the users’ attitudes and drinking behaviours, and the potential determinants.

## 2. Materials and Methods

Tap water in Germany undergoes regular checks and is safe for drinking. Authorities also regulate the number of sanitary facilities in schools, including an adequate number of mostly flat taps with sinks for hand washing (which are inconvenient for bottle filling), depending on the number of students. The two drinking fountains installed in our study setting are water taps without bubblers (adding water without carbon dioxide), located on the schools’ floors. Both fountains fill bottles placed under a sensor. The Hauptschule’s fountain also produces a small, curved jet of water for drinking directly, activated by a button (see Appendix A). Apart from drinking water from taps and via the two fountains, the two schools do not offer free drinks. Instead, cafeterias in both schools sell a variety of drinks, including spritzers and bottled water. Students in the Hauptschule and the Gymnasium have a similar socioeconomic composition, according to NRW’s social index.

The study has a descriptive, cross-sectional design using questionnaires. The questionnaires contain 26 closed and two open items for students (Appendix A) and 17 closed and two open items for teachers (Appendix A), respectively. All students in years 6, 8, and 10 and their class teachers were invited to participate and were provided with the study information and consent forms. All children who returned their caregivers’ informed consent completed the survey in class.

The questionnaire’s content resulted from literature research, consultation with the schools’ principals, and views from child health experts. A small group of children assessed the questionnaire for understandability. Apart from sociodemographic items, the questionnaires included items on water fountain use and potential determinants of drinking behaviours, such as the availability of alternatives, prior experiences, flavour preferences, the role of peers, and aspects of health literacy. The open-ended questions allowed commenting on, for example, the appearance and function of the water fountains.

Insufficient German language skills and status as a guest student were exclusion criteria. Responses were transcribed from paper records to an MS Excel file for descriptive analysis, and 10% of the data entered were crosschecked to identify transcription errors. In addition, one researcher conducted two 15 min semi-structured interviews online via Zoom^R^ version 5.9.1 with the principals of the two schools following a topic guide (Appendix A). Free text responses to the open questions in the questionnaires and the records of the principals’ interviews were analysed by content analysis using MAXQDA^R^2022 software [22,23]. Quantitative data were analysed using SPSS^R^ version 28 software. Logistic regression served to identify risk factors for using the drinking fountain and students’ attitudes with responses to “Do you use the drinking fountain regularly?” and “Drinking fountains are a good idea” as dependent variables. A *p*-value of < 0.05 was considered statistically significant.

## 3. Results

The response rate for students’ questionnaires was 36.1% (162/448). The best rates had students of the Hauptschule (63%) compared to students of the Gymnasium (31.1%) and teachers 43.5% (10/23). The analysis included all 162 questionnaires returned by the students (Figure 1). However, not all students answered all questions. Therefore, in the following, the proportions’ denominators in brackets refer to only the number of provided answers, excluding missing responses.

Of all students, 53.1% (*n* = 86) were female, 39.5% (*n* = 64) were male, 6.2% (*n* = 10) indicated diverse, and one participant each preferred not to answer this question or left the answer open (Table 1). The median (range) age of all responding students was 13 (11–18) years, and nearly half of the responding students (*n* = 78) were in year 6.

### 3.1. Students’ Responses

#### 3.1.1. Use of the Drinking Fountains and Drinking Behaviour

More than half (55.7%; *n* = 88) of all students responded that they used the drinking fountain regularly (Figure 2). However, bringing drinks from home was common (87% agreed, *n* = 138), and many children denied bringing fewer drinks from home since the drinking fountains were put into operation (78.9%, *n* = 127). In addition, 39.8% (*n* = 47) used the fountains only less than once, and 26.3% (*n* = 31) used them one to two times per week, while 31.1% of respondents (*n* = 50) reported regularly purchasing drinks. A fifth of respondents (*n* = 33) knew about drinking fountains from their primary schools.

Whether brought from home (Figure 3) or purchased (Figure 4), water was the preferred option of all drinks in 63% (*n* = 136) and 31% (*n* = 46) of respondents, respectively, although juice spritzers were also popular (reportedly bought by 23%, *n* = 34).

#### 3.1.2. Acceptance of the Drinking Fountains

Most students found drinking fountains at school a good idea, recommended them to other schools, and felt able to distinguish between healthy and unhealthy drinks (complete agreement by 73.3% (*n* = 118), 73.1% (*n* = 117), and 70.5% (*n* = 110), respectively). However, only 26.5% (*n* = 43) appear to consume more water since installing the drinking fountains (Figure 5).

Responses to the question about considering healthiness when selecting drinks, fellow students’ use of the drinking fountain, and coverage of healthy drinks as topics in the classroom were less definite (complete agreement by 19.8% (*n* = 32), 25.5% (*n* = 41), and 30.1% (*n* = 47), respectively) (Figure 5). However, 94.9% (*n* = 150) of respondents described water as “healthy”, while juice spritzers were considered “somewhat healthy” by 51.6% (*n* = 81) and “somewhat not healthy” by 38.9% (*n* = 61).

There were only marginal differences in students’ responses between the two schools to the questions on attitudes towards the drinking fountain, regular use, and drinking more water since the drinking fountains have been in place (Appendix A).

Univariate ordinal logistic regression analyses did not reveal significant risk factors for students’ use of (Appendix A) and attitudes (Appendix A) towards drinking fountains.

#### 3.1.3. Students’ Responses to Open Questions

Seventy-four percent (120/162) of students used the questionnaire’s free text option to suggest changes to the drinking fountains (multiple answers). About half of all answers recorded referred to the drinks offered (50.8%; *n* = 61), whereby the most frequently expressed wish was for sparkling water (28.3%, *n* = 34). Other wishes of the students related to the fountains’ design, alternative ways of accessing water (e.g., by supplying cups), or a simpler operation. Further comments were made by 38.9% of the respondents (*n* = 63), whereby 30.2% (*n* = 19) of the entries welcomed the drinking fountains project.

### 3.2. Teachers’ and Principals’ Responses

The ten teachers included in the questionnaires’ analysis agreed with the statement “Drinking fountains are a good idea” either fully (*n* = 8) or partly (*n* = 2). However, out of the ten, only four indicated using the fountains regularly, and out of those two, “less frequently than once a week”; the other two did not answer (Appendix A). Although five agreed entirely and one partially to have dealt with drinking fountains in the classroom, one would rather not agree, and three had not covered the topic.

During the short, semi-structured interviews, the two principals highlighted the provision of water by fountains anytime at no cost and the reduced consumption of sweetened drinks and waste caused by empty plastic bottles. Principals considered several aspects of planning for and running the fountains. For example, alignment to ensure fountains are easily accessible and to fit the school buildings’ architecture and construction consumed time and required considerable organisational skills, especially under the public health measures set up to prevent SARS-CoV-2 infections. Principals felt meeting the fountains’ maintenance and repair cost was more complicated than the initial installation. They viewed all stakeholders’ understanding and acceptance of drinking fountains as paramount for their success. They recommended involving parents, colleagues, and students at an early planning stage. To achieve such involvement, principals found communication via social media and the health department’s provision of recyclable water bottles helpful.

## 4. Discussion

In our cross-sectional study among 162 students from years 6 to 10, 94.9% view water as a healthy choice, 87% of the students report bringing drinks to school regularly, preferably water, and confidently distinguish healthy from unhealthy drinks. However, about three-quarters of respondents keep obtaining drinks from home, and only 26.5% have drunk more water since the fountain’s installation. In addition, despite more than 70% of students finding drinking fountains a good idea, only 55.7% appear to use them regularly; two-thirds reported using them once or twice a week or less, and 31.1% regularly buy drinks at school kiosks. The ten teachers surveyed and the schools’ two principals viewed the fountains unanimously positively, highlighting the advantages of no cost for students, ease of access and waste avoidance, and being aware of challenges during planning and meeting running costs.

Our pilot study reveals a disconnect between good knowledge of drinking water as a healthy choice and a generally good attitude towards drinking fountains among students and the fountains’ use by students. The findings may fuel the debate about what determines students’ drinking choices in schools, especially at the secondary level. For example, almost all respondents in our study felt the drinking fountain was a good idea and that other schools should also have one. However, only slightly more than half of the students use it regularly, and 71% voted “not” or “not likely” to drink more water since installing the drinking fountains. In addition, for most students, the amount of water they brought from home remained unchanged after the fountains’ installation. These findings seem to contrast with a randomised German study of 2950 primary school students in years 3 and 4 [7]. Installing drinking fountains increased water consumption by 1.1 glasses per student and reduced the rate of obesity by 31% compared to the control group. However, participants in our pilot study came from secondary schools, and their drinking behaviours will likely differ.

Different health determinants might also be responsible for other conflicting findings. For example, in the US, the availability of drinking fountains in school cafeterias was positively associated with water consumption, mostly during lunchtime [24]. Hence, accessibility and convenience appear to be important for water consumption. Conversely, in our study, each school had only one fountain, a potentially critical reason for limited use. However, data also show that drinking fountains performed poorly compared to other ways of providing water (e.g., bottled water) [25]. In addition, younger students were more likely to use available water sources than older students, which may underscore the importance of age-dependent determinants for healthy drinking choices. In our study, year 6 students were the most likely to use the fountain. However, they were the only group who received reusable water bottles as part of the “Iss ok” project. Apart from country differences and age, incentives, such as free water bottles or means that may facilitate and ease drinking from the tap, e.g., by issuing cups, may increase the acceptance and use of drinking fountains [26].

Indeed, practical aspects may influence students’ choices for or against using the drinking fountain. For example, a more attractive design may have a positive effect [27] that could mitigate students’ rejections of drinking fountains and thus water consumption “from the tap” [28]. In addition, students often consider drinking fountains in schools unsafe or even pathogenic, according to US studies [29], perhaps explaining the discrepancy between attitudes to water as a healthy choice and the use of drinking fountains observed in our study. Finally, students’ desire for carbonated (sparkling) water coming out of the fountains and that 64% report bringing water from home may support the hypothesis that aesthetic, hygienic, and practical considerations play a role in students’ choice of water sources for drinking.

The school context provides the opportunity to promote water drinking through drinking fountains and to address misconceptions and false assumptions through teaching and learning. However, our findings suggest room for improvement. For example, less than half of the students remembered that drinking water or drinking fountains were covered in class; from the teachers’ perspective, six of ten reportedly had dealt with the subject in class and admitted not using the drinking fountains. Teachers could be more aware of their function as role models and could consider more targeted use of age-appropriate teaching materials with shown benefits [7,26].

Our study has limitations. The study is observational and has no control group. The descriptive, cross-sectional design provides only a snapshot of the situation at the two schools and does not provide evidence of causality. A before-and-after comparison would be more appropriate to estimate the effect of the intervention. In addition, external factors, such as the SARS-CoV-2-related public health measures at the school and the relatively cool temperatures during the survey (winter), will further limit the generalisability. Nevertheless, the cross-sectional design and the mix of quantitative and qualitative methods meet the aim and objectives of this explorative study and may trigger a more in-depth investigation. Furthermore, the study represents real-life data that add value to data collected under controlled conditions without biases caused by incentives and additional resources. Its findings and conclusions generated hypotheses about the acceptability and users’ behaviours that may result in more extensive studies with different designs to test them.

In summary, most students in our study value and are aware of water as a healthy drinking choice and approve of drinking fountains in principle. However, only a fraction of students use the fountains regularly, often infrequently, and the availability of only one fountain per school limits access. The observed discrepancy between attitude and use observed in students might hold for teachers, too. Principals stress the fountains’ advantages, costs, and organisational challenges. Enhanced involvement of teachers in drinking fountain promotion and systematic engagement of students with water drinking opportunities for a healthy diet may increase students’ use and acceptance.

## Figures and Tables

**Figure 1 children-10-00817-f001:**
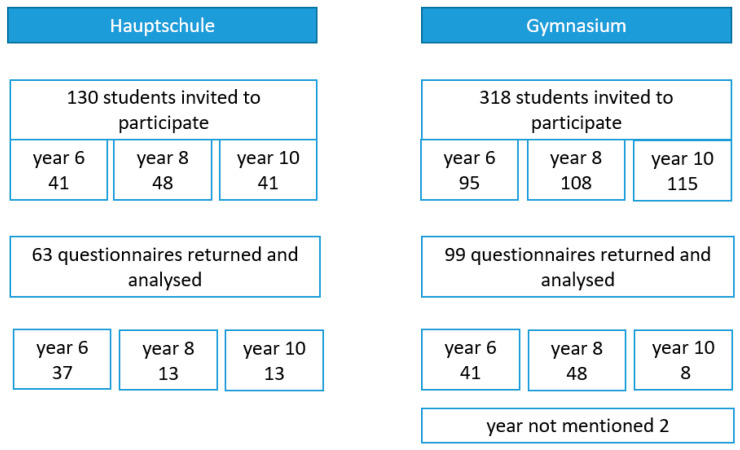
Study profile.

**Figure 2 children-10-00817-f002:**
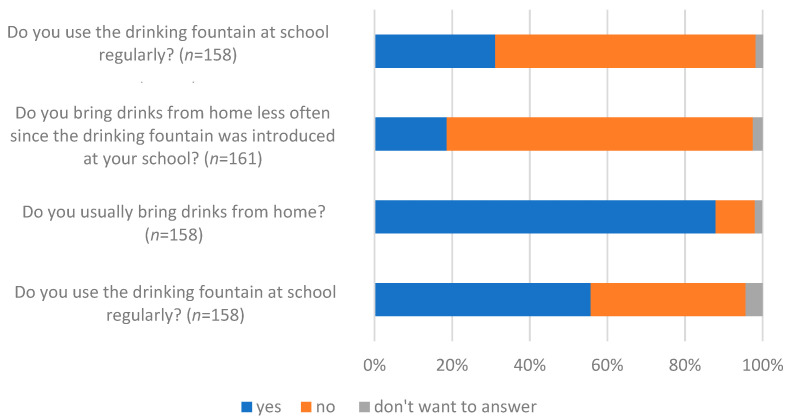
Use of drinking fountains among students at both schools (*n* indicates the number of responses).

**Figure 3 children-10-00817-f003:**
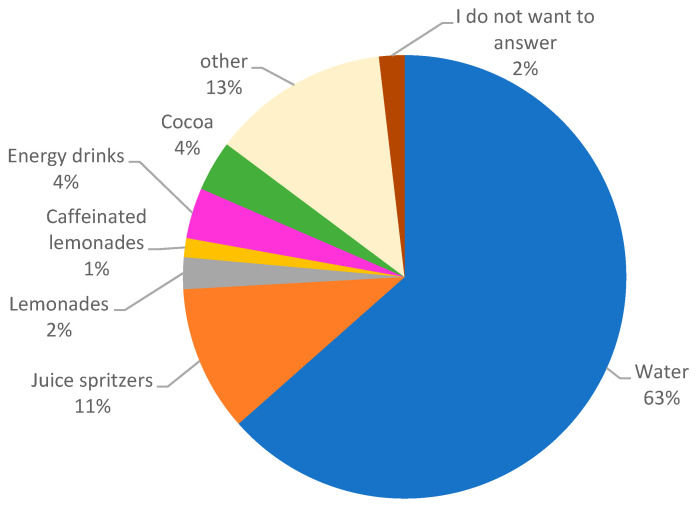
Drinks brought from home by students at both schools (number of answers *n* = 216), multiple answers.

**Figure 4 children-10-00817-f004:**
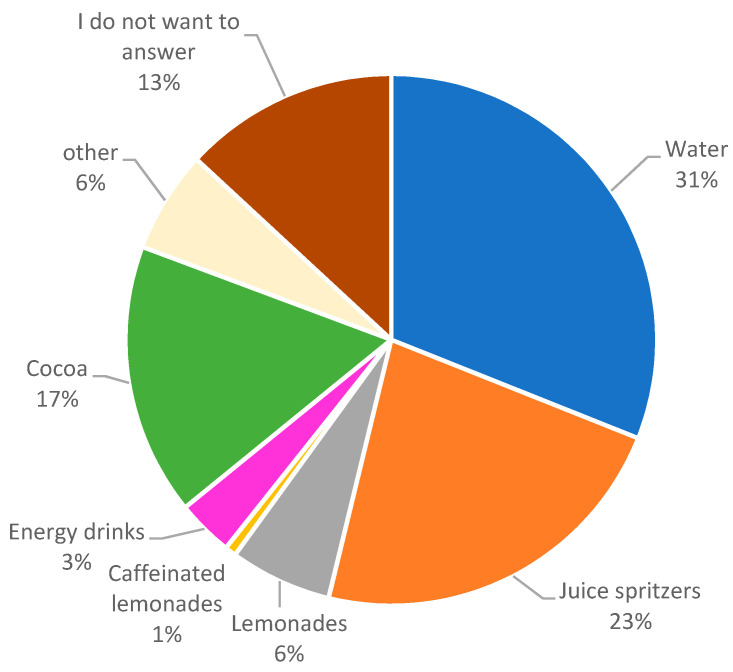
Drinks purchased by students at both schools (number of answers *n* = 146), multiple answers.

**Figure 5 children-10-00817-f005:**
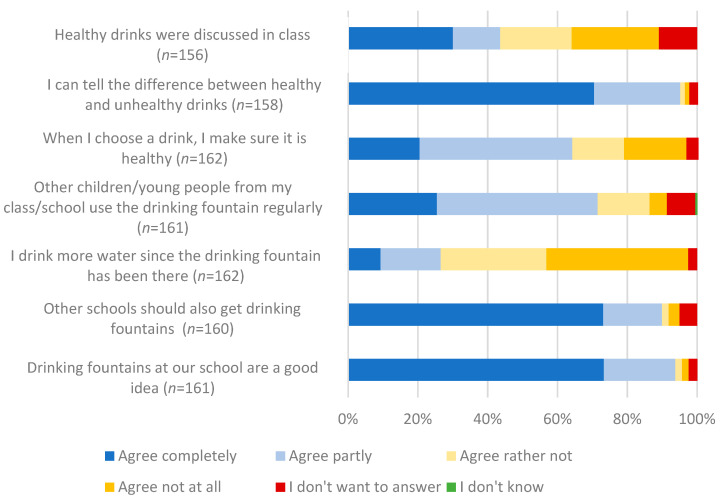
Acceptance of drinking fountains among students at both schools (*n* indicates the number of responses).

**Table 1 children-10-00817-t001:** Student demographics (*n* = 162).

	Quantity	%
Gender		
Female	86	53.1
Male	64	39.5
Diverse	10	6.2
Not specified	1	0.6
Missing	1	0.6
School type		
“Hauptschule” *	62	38.3
“Gymnasium” ^§^	98	60.5
Not specified	2	1.2
Years in school		
Year 6	78	48.1
Year 8	60	37.0
Year 10	21	13.0
Not specified	3	1.9

* There is no exact equivalent for the German “Hauptschule” in the British school system. Therefore, the term “general secondary school” may best describe this secondary school type. ^§^ There is also no exact equivalent for the German “Gymnasium”. The UK’s “Grammar school” might be the most appropriate equivalent.

## Data Availability

The quantitative data presented in this study are available upon request from the corresponding author. However, the qualitative data are not publicly available due to privacy and ethical considerations.

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
