# Peer review of "Use and Acceptance of Drinking Fountains: A Pilot Study in Two Secondary Schools in Dortmund, Germany"

_children, 2023, doi:10.3390/children10050817_

Round 1

Reviewer 1 Report

Thank you very much allow me to review the article “Brief Report” entitled “Use and acceptance of drinking fountains: a pilot study in two secondary schools in Dortmund, Germany.” (children-2347300).  This is an exploratory study assesses the use and acceptance of drinking fountains in Dortmund's schools (municipal "Hauptschule") in two secondary schools addressing students and teachers. It describes the frequency of the fountains' use, the users' attitudes and drinking behaviors, and the potential determinants.  Comments:  The first sentence about the importance of drinking enough water in children's health should be used as a bibliographical reference. In the introduction, a little more should be collected about the habit of drinking water among young people in the area where they are going to study or about the knowledge that they have about water consumption in other studies, this would allow a better assessment of the contribution of the work. presented. in material and methods, I suggest that it be incorporated The type of design used, the number of children who were offered to participate in the study, the participation rate, the approval of the ethics committee that allowed the study to be carried out, the calculation of the size of the sample and report whether they are validated questionnaires or not. In results, the basic characteristics of the total are presented, but it is not separated by consumption or non-consumption, as would be expected according to the objective. the discussion should consider that the previous water consumption and the one after the incorporation of the sources are not evaluated, a design before and after would be the ideal objective presented.

Reviewer 2 Report

This descriptive study of student beverage choices was of some interest and I appreciate the opportunity to provide review. This fairly small study in two intervention secondary schools (note:  school terminology in paper was confused; one was described as a “grammar school;” at least in the US this would imply an elementary school), with no control schools, describes student and staff attitudes to drinking fountains and student behavior in terms of fountain use, beverages consumed and source of beverages consumed.

Background questions that need answering, particularly for the paper to be helpful to those outside of Germany:

·      Is there any regulation requiring access to drinking water in schools, e.g., plumbing codes requiring a certain density of drinking water outlets, or other policy related to water access?  The paper makes it sound as if the study schools previously had no no-charge drinking water.

·      It is critical to describe the type of drinking fountains being discussed, i.e., “traditional fountain" (“bubbler”), water bottle filler, combination unit (bubbler plus bottle filler) or other.

·      It would be valuable to know what types of drinks are available for purchase at school and if any drinks are provided at no charge as part of a school meal service, and whether any school or state policies bear on this.

Other considerations

·      Other factors can influence use of school drinking water dispensers, e.g., perception of tap water safety, accessibility and appeal (e.g., cleanliness and appearance) of the water dispenser, and drinking water education and promotion in the school (Muckelbauer et al., Kenney et al.).  While this is discussed (l 208-217), a literature review prior to determining study methodology should have suggested that these variables should have been included in the study survey instruments.

·      Were differences seen between the two schools? What are differences (e.g., socio-economic) between the two student bodies?

·      In general I would like to see a more complete review of the literature

·      The intervention appeared to consist of one (1) drinking fountain per school. It should be clarified whether this is the only access to no-charge drinking water. If so, this may be the critical reason why, despite understanding the benefits of water and being favorable about school drinking fountains, the presence of fountain did not much change behaviors.  Yet this factor is ignored in the study - or clarification is needed.

L 34 -35  Specify that replacing caloric beverages with water (assuming no other change in diet) can prevent obesity.

L 199  With reference to statement about citation 13, it should be noted that the study looked specifically at presence of a drinking water dispenser in the school cafeteria.  All US schools are subject to state plumbing code requiring a certain density of water fountains.  Therefore the sentence “children from schools with drinking fountains drank more water than children without” is misleading.

If possible, please translate the supplementary materials also. (I could read the German but most would not be able to).

Round 2

Reviewer 1 Report

I have carefully reviewed the new version of the manuscript article "Brief Report" entitled "Use and acceptance of drinking fountains: a pilot study in two secondary schools in Dortmund, Germany" (children-2347300). (children-2347300) as well as the authors' response to the comments made.

The authors have correctly answered the questions and suggestions made and have improved the manuscript.

However, if the aim is to assess the consumption of water once it has been placed in fountains in schools, it is an assessment of the prevalence of water consumption from fountains. This implies that the sample size can be calculated on the basis of the estimated prevalence. I do not agree with the authors that it cannot be calculated. On the other hand, in the table you can see whether or not there are significant differences in the distribution of proportions with a simple test of comparison of proportions.

Considering that this is a pilot study we can evaluate the information provided in the paper. I hope that the recommendations made will be useful for future work.

Reviewer 2 Report

I approve of the revisions to the manuscript.  Thank you.